# Microstructure Characteristics and Mechanical Properties of Flash Butt Welded 590 MPa V-N Microalloyed Heavy-Duty Truck Wheel Steel

**Cairu Gao [1], Kaiyu Cui [2], Huifang Lan [1], Tao Liu [3], Linxiu Du [1,*], Yujiao Ma [4], Xinxin Guo [4] and Chenshuo Cui [4]**

[1] State Key Laboratory of Rolling and Automation, Northeastern University, Shenyang 110819, China
[2] State Key Laboratory for Comprehensive Utilization of Vanadium and Titanium Resources of Panzhihua Iron and Steel Group, Panzhihua 617000, China
[3] New Materials Research Institute of Nanjing Iron and Steel Co., Ltd., Nanjing 210000, China
[4] Changshu Institute of Technology, College of Automotive Engineering, Suzhou 215500, China
* Correspondence: dulinxiu2000@163.com or dulx@ral.neu.edu.cn; Tel.: +86-024-8368-6412

**Abstract:** This study reports the welded joint of a novel 590 MPa V-N microalloyed wheel steel on microstructure and mechanical properties after flash butt welding. The welding parameters were flash current 48°/582.0 A, upsetting current 44°/516.6 A, and workpiece clearance 1.5 mm. The evolution of microstructure in the welded joint occurred as follows: welding seam (ferrite side plate + acicular ferrite +martensite)→coarse-grained zone (acicular ferrite + granular bainite)→fine-grained zone (fine-grained ferrite + M/A island)→base metal (equiaxed ferrite + pearlite). The standard impact energy value of welding seam, coarse grain zone, fine grain zone, and base metal at −40 °C was 116, 128, 144, and 88 J, respectively. The mechanical property of the joint was excellent. The microstructure, the number of grain boundaries, and the dislocation density directly affected the strength and hardness of the joint. The increase of large angle grain boundaries and the decrease of effective grain size were beneficial to the improvement of toughness. The hot-rolled 590 MPa V-N microalloyed wheel steel had superior weldability.

**Keywords:** wheel steel; weldability; microstructure; mechanical property

## 1. Introduction

The application of lightweight materials is inseparable from the development of automobile lightweight. The material, structure, and quality of wheels are not only related to driving safety, but also have an important impact on vehicle servicing, ride comfort, and energy saving [1,2]. As the key component of the vehicle driving system, wheel steels are widely used with the rapid development of the automobile industry. Overall, 330CL, 420CL, 440CL, etc., with large consumables and low strength and performance still account for a large proportion of wheel steel in China. Microalloying with Nb and Ti in combination is widely adopted in wheel steels, but there is no V-N microalloying system at present [3]. Compared to the Nb-Ti microalloyed steel, V-N steels exhibit grain refinement through intragranular nucleation of ferrite on VN precipitates, partly because of low lattice mismatch of VN with ferrite. The addition of N in V microalloyed steels stimulates the precipitation of V carbonitrides and increases their volume fraction. The insoluble MnS in the steel provides heterogeneous sites for nucleation of VN and the strength and toughness of welded joints are simultaneously improved by nucleation of acicular ferrite on MnS + VN complex inclusions [4,5]. The price of microalloying elements fluctuates greatly in the market, so it is necessary to develop new high-strength wheel steels as a technical reserve in the automobile industry, and the new steel developed for wheels must meet the requirements of mechanical and welding properties.

Flash butt welding is a widely used welding method in rim production. Flash butt welding is resistance pressure welding. It can not only weld the compact surface but can

also weld the weldment with expanded section. It has high automation, good welding quality, high welding efficiency, and many kinds of weldable metals. Automobiles have high requirements for the formability and strength balance of the materials used for making wheels. The formability of wheel steel mainly refers to the cold bending performance and the hole expanding performance. Cold bending performance is an important index to measure the formability of wheel steel, which directly affects the product quality and economic benefits. Ichiyama [6,7] systematically studied the cold bending performance and impact toughness of flash butt welded joints. The bending and impact tests of the joint were carried out under different process parameters. The results showed that the cold bending performance of the joint became worse with the increase of preheating time and upsetting allowance within a certain range. The oxide inclusions in the impact fracture of the joint are mainly composed of O, Si, Mn, and Al. Ziemian [8] studied the effect of different flash butt welding process parameters on the microstructure and properties of ASTMA529 carbon manganese steel joint. The results showed that the joint microstructure was acicular ferrite, side plate ferrite, and widmansite ferrite. The maximum hardness occurs at the weld, the HAZ softens, and the defects and oxide inclusions on the joint end face are related to the upset allowance.

It is well-known that wheel rims are made from a rectangular sheet metal by welding, flaring, and rolling. As a typical resistance welding, flash butt welding is widely applied in the forming of wheel rims because of the advantages of low cost, high welding efficiency, and speed [9–11]. Due to the effect of welding thermal cycle, there are obvious differences in microstructure and mechanical properties between the welding seam, heat-affected zone (HAZ), and base metal, especially in the softening behavior that may occur in HAZ, which seriously affects the service life of rims and the safety of wheels during service [12,13]. In order to ensure good quality of rims, it is essential to study the microstructure and mechanical properties of welded joints.

The research focuses on microstructure characteristics and strengthening mechanism of 590 MPa high strength wheel steel. The microstructure, hardness distribution, strength, and toughness of flash butt welding joint were analyzed. The results provide a necessary theoretical basis for the application of 590 MPa high strength wheel steel for heavy-duty trucks.

## 2. Experimental Procedure

### 2.1. Materials

Table 1 shows the chemical composition of the experimental steel. A proper amount of Si and Mn elements were added to the low carbon steel, and the precipitation strengthening was promoted through V and N microalloying so as to improve the strength and toughness. The steel was melted in a vacuum induction furnace and a 50 kg ingot was produced. Controlled rolling and controlled cooling experiments have been carried out on experimental steel. The homogenization of the ingot was carried out at 1200·°C for 2 h. The finishing rolling temperature has a great influence on the mechanical properties. A temperature that is too high will lead to grain coarsening, and elongation decreases due to temperatures that are too low, considering the on-site rolling rhythm and temperature drop. The finishing temperature and coiling temperature are 780–830 °C and 450–550 °C, respectively. Tensile tests based on standard of ISO GB/T 228.1-2010 were conducted at room temperature using a SANS-5105 tensile testing machine. The impact test was conducted on a 250 HV fully digital instrumented pendulum impact tester using standard V-notch Charpy impact specimens (10 × 10 × 55 mm) according to ASTM A370.

**Table 1.** Chemical compositions of experimental steel (wt.%).

| C | Si | Mn | P | S | Al | V | N | Fe |
|---|----|----|---|---|----|---|---|----|
| 0.11 | 0.13 | 1.52 | 0.006 | 0.001 | 0.036 | 0.092 | 0.0127 | Bal. |

The equivalent carbon content ($C_{eq}$) and the welding crack susceptibility index ($P_{cm}$) were 0.4 and 0.2, respectively, calculated by Equations (1) and (2) [14,15].

$$C_{eq} = C + \frac{Mn + Si}{6} + \frac{Ni + Cu}{15} + \frac{Cr + Mo + V}{5} \tag{1}$$

$$P_{cm} = C + \frac{Si}{30} + \frac{Mn + Cu + Cr}{20} + \frac{Ni}{60} + \frac{Mo}{15} + \frac{V}{10} + 5B \tag{2}$$

*2.2. Methods*

The experimental steel plates were connected by flash butt welding and the welding process was conducted under the following conditions: flash current 48°/582.0 A, upset current 44°/516.6 A, and a workpiece gap of 1.5 mm. The microstructure of the joint was examined by OLYMPUS optical microscopy (OM), FEI Quanta 600 scanning electron microscopy (SEM), and FEI Tecnai $G^2F20$ transmission electron microscopy (TEM). The grain orientation was evaluated by electron backscattered diffraction (EBSD). Tensile tests were performed by the WDW-300 tensile test machine. The hardness of the joint was determined by a hardness tester with a load force of 10 N and a hold time of 15 s. Low temperature impact toughness was conducted by INSTRON 9250 drop hammer impact test machine at a temperature of −40 °C.

## 3. Results

*3.1. Welding Process Parameters*

Through a series of process tests, the weldable range of vanadium microalloyed wheel steel was explored. The welding defects of the test steel joint are shown in Figure 1. When the flash current and upset current are lower than 42°, the joint will produce incomplete fusion defects (Figure 1a). The flash current is too small and the heat input is too small, and the metal at the weld is not melted enough to conduct upsetting and cooling, resulting in incomplete fusion. When the flash current is higher than 70° or the upset current is higher than 60°, the melting phenomenon will occur at the joint, resulting in the inability to weld (Figure 1b). When the flash current is too large, the excessive heat input will melt the metal at the weld. The weldment cannot be welded, and the metal will melt. According to the process test results, the weldable range of the test steel is: flash current 42~70°/484.3~946.4 A, upset current 42~60°/484.3~781.6 A. The weldability of the tested steel is good within the weldability range.

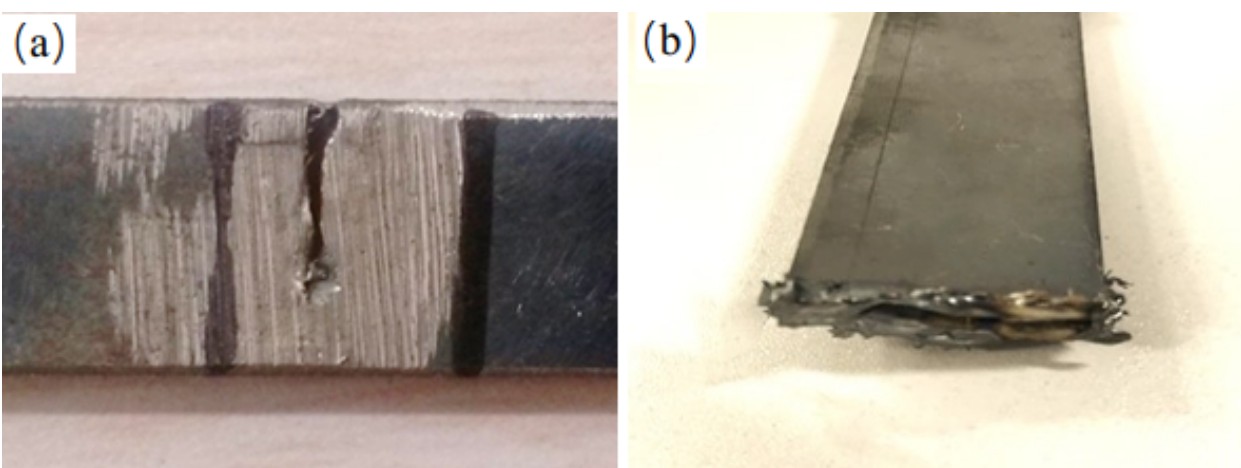

**Figure 1.** Defects in the welded joint of test steels. (**a**) Incomplete Fusion Defects caused by low current; (**b**) Melting defect caused by excessive current.

The optimal value of fixed flash current is 48°, and the upsetting current is 42°, 44°, 46°, and 48°, respectively. The influence of different upsetting current on joint reinforcement and HAZ width was studied. The joint reinforcement and HAZ width under different upset currents are shown in Figure 2. At 48° flash current, when the upset current is 42~48°, the variation range of joint reinforcement is 2.33~3.83 mm. At the same time, the joint reinforcement is greatly affected by upset current and increases with the increase of upset current. The influence of upsetting current on the HAZ width of the joint is relatively small. When the upsetting current is 44°, the HAZ width of the joint is the smallest, at 4.89 mm.

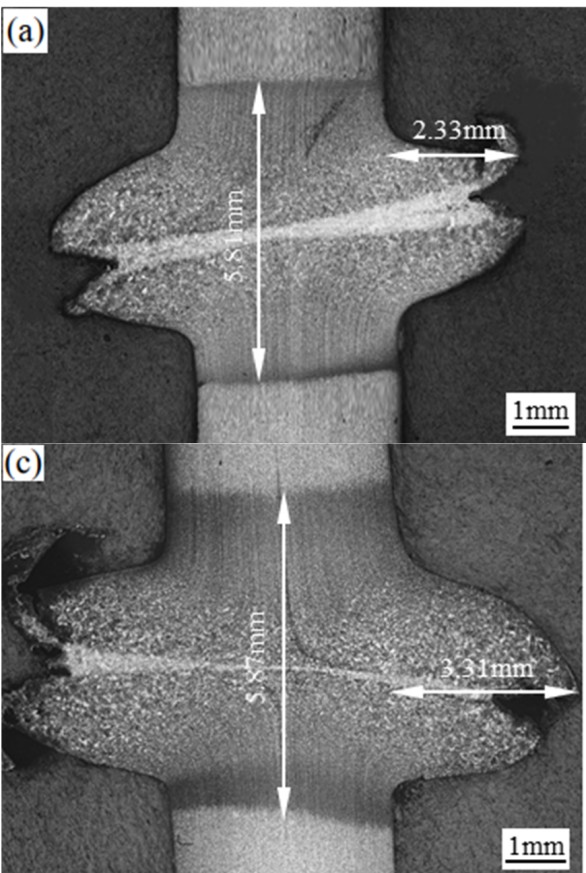
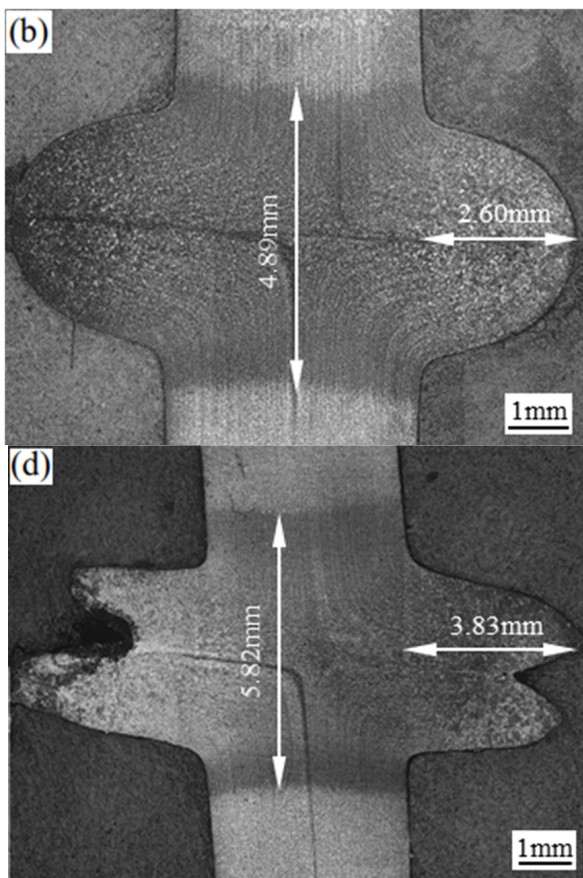

**Figure 2.** The residual height and HAZ width of the joint under different upset current. (**a**) 42°; (**b**) 44°; (**c**) 46°; (**d**) 48°.

Combining the joint reinforcement and HAZ width under different upsetting currents, it can be found that the joint reinforcement and HAZ width are relatively small when the upsetting current is 44°. In order to determine the optimal value of upsetting current more accurately, the tensile properties of the joint under 48° flash current are 42°, 44°, 46°, and 48°, respectively. The tensile results of the joint under different upsetting current are listed in Table 2. Under the flash current of 48° and upset current of 44°, 46°, and 48°, the tensile specimen of the joint breaks at the base metal. When the upset current is 42° (close to the lower limit of the weldable range), the joint breaks at the weld and the elongation is only 11.3%. Since the joint will break at the weld when the upsetting current is 42°, the optimal process parameter value of upsetting current should be between 44° and 48°.

The fixed flash current is 48°. The hardness distribution of the joint is studied when the upset current is 44°, 46°, and 48°. The hardness distribution of the joint under different upsetting current is shown in Figure 3. Similar to the joint hardness distribution under different flash current, the joint hardness gradually decreases from weld to base metal under different upset current; When the upsetting current is 44°, the average hardness value of the joint area is the highest and the maximum hardness value occurs at the weld,

which is 284.9 HV. According to the results of joint hardness distribution under different upsetting currents, the optimal value of the upsetting current was determined as 44°.

**Table 2.** The result of the tensile test of the joint under different upset current.

| Number | Flash Current/° | Upset Current/° | $R_{eL}$/MPa | $R_m$/MPa | $A_{50}$/% | Fracture Position |
|---|---|---|---|---|---|---|
| 1 | 48 | 42 | 559.3 | 649 | 11.3 | weld seam |
| 2 | 48 | 44 | 560.7 | 651 | 22.7 | Base metal |
| 3 | 48 | 46 | 557.3 | 657.3 | 21 | Base metal |
| 4 | 48 | 48 | 568 | 660.3 | 20 | Base metal |

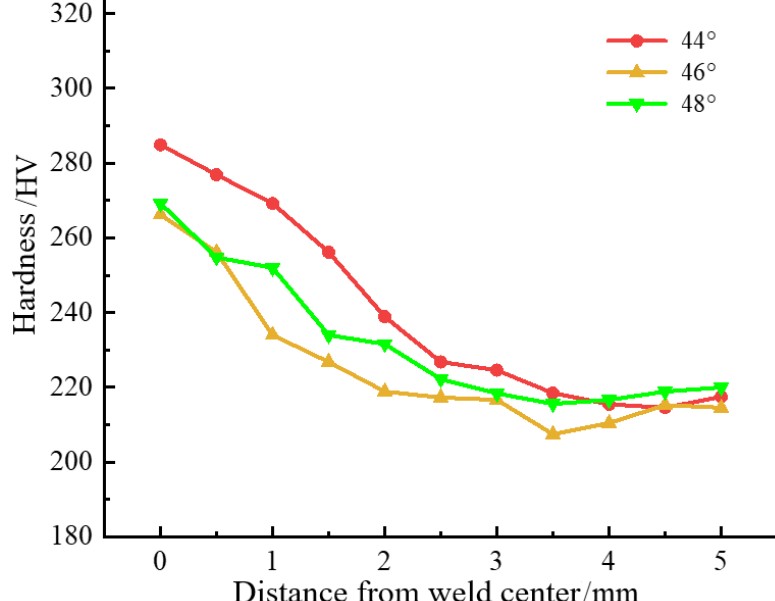

**Figure 3.** Hardness distribution of the joint under a different upset current.

To sum up, the optimal process parameters for flash butt welding of the test steel are flash current 48°/582.0 A and upset current 44°/516.6 A. Under this welding process, the joint reinforcement is 2.60 mm, the heat affected zone width is 4.89 mm, and the comprehensive mechanical properties are excellent.

*3.2. Microstructure of Welded Joint*

The welded joint is divided into the following parts: welding seam (WS), coarse grain zone (CGZ), fine grain zone (FGZ), and base metal (BM). Figure 4 shows the microstructure of different regions of the welded joint.

The microstructural constituent of WS (Figure 4a,b) can be observed to be composed of ferrite-side-plate (FSP), acicular ferrite (AF), and martensite (M). The microstructure contains more lath martensite and a small amount of coarse bainite ferrite (BF) grains. The distribution of martensite laths is fine and uniform. The microstructure of CGZ (Figure 4c,d) contains acicular ferrite (AF) and granular bainite (GB). GB has a high content and is evenly distributed, and AF is island distributed. The short rod and block M/A islands are distributed dispersedly. The microstructure of FGZ (Figure 4e,f) is comprised of fine grain ferrite and a certain amount of M/A islands. M/A islands are characterized by significant banded distribution. Banded structure can cause anisotropy of steel and make the plasticity and toughness of steel worse. Ferrite presents massive and discontinuous morphology characteristics. The microstructure of BM (Figure 4g,h) consists of equiaxed ferrite and pearlite and it has a banded distribution along the rolling direction.

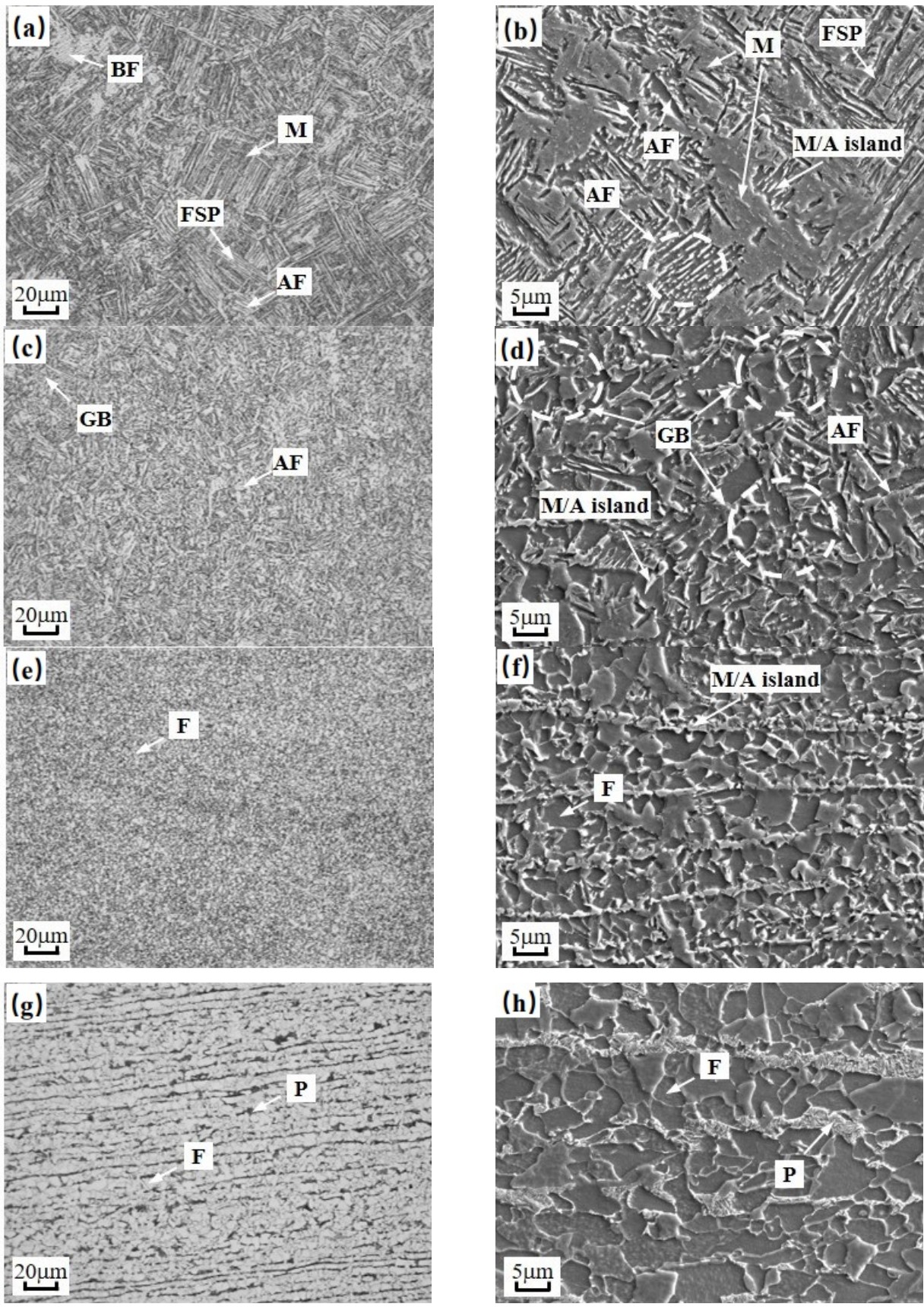

**Figure 4.** OM micrographs and SEM micrographs of different regions of the welded joint. (**a**,**b**) WS; (**c**,**d**) CGZ; (**e**,**f**) FGZ; (**g**,**h**) BM.

### 3.3. Mechanical Property of Welded Joint

Table 3 shows the results of tensile test for flash-welded joint. Based on tensile test results, the fracture position is in BM and the average tensile strength and elongation were 651 MPa and 20.7%, respectively.

**Table 3.** Results of tensile test for flash-welded joint.

| Number | $R_{eL}$/MPa | $R_m$/MPa | $A_{50}$/% | Fracture Position |
|--------|--------------|-----------|------------|-------------------|
| 1 | 559 | 646 | 21 | BM |
| 2 | 566 | 654 | 20 | BM |
| 3 | 557 | 653 | 21 | BM |

The hardness distribution of the welded joint is shown in Figure 5. Compared with BM, the hardness of WS and HAZ was increased to a certain extent, and the increase in WS was most obvious. The average hardness values of WS, CGZ, FGZ, and BM were 280.9, 262.7, 232.9, and 217.3 HV, respectively. In terms of the hardness distribution of the welded joint, there is no softening tendency in the welded joint.

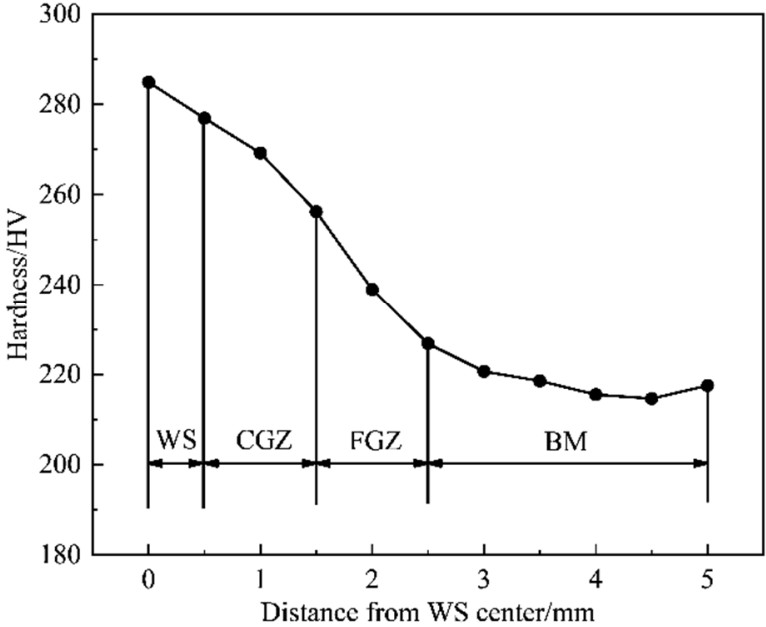

**Figure 5.** Hardness distribution of the welded joint.

The standard impact energy value at −40 °C of WS, CGZ, FGZ, and BM was 116, 128, 144, and 88 J, respectively. Figure 6 shows the fracture morphology of different regions of the welded joint. The fracture mode of each region was ductile fracture. The dimples in BM were relatively large and shallow, somewhat smaller and deeper in WS and CGZ, and the dimples in FGZ are the deepest compared with the other three regions. Isoaxial dimples were distributed in the impact fracture morphology of different regions of the joint, and no fluvial cleavage fracture feature was found. Therefore, it can be determined that the fracture mode in each region belongs to ductile fracture. Cavities or coarse particles (Figure 6d) fully expand and polymerize under tensile and shear stresses, resulting in fracture, presenting equiaxed dimples of relatively uniform size. The final separation occurs along the direction of the maximum shear force, resulting in an overall shear fracture.

In the process of wheel rim forming, many processes involve bending, so it is necessary to evaluate the cold bending formability of the joint under the optimal welding process. According to the actual needs, the bending angle is generally less than 90° in the actual bending forming process of the rim, so it is necessary to evaluate the cold bending

performance of the joint under the optimal welding process parameters when $\alpha = 90°$. The surface state of the cold-formed joint specimen is shown in Figure 7. It can be observed that when $\alpha = 90°$, there are no macroscopic cracks visible on the surface and side of the weldment, which indicates that the cold bending performance of the joint under the optimal welding process is qualified.

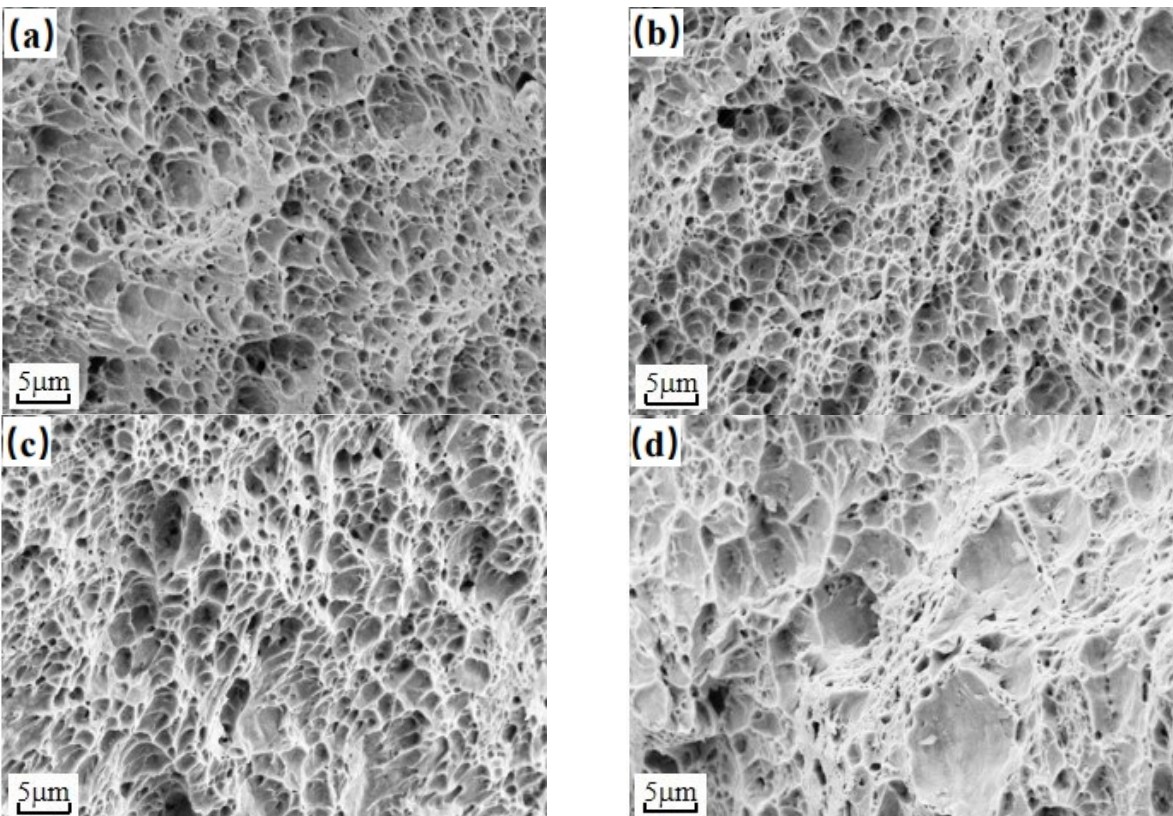

**Figure 6.** Fracture morphology of different regions of the welded joint. (**a**) WS; (**b**) CGZ; (**c**) FGZ; (**d**) BM.

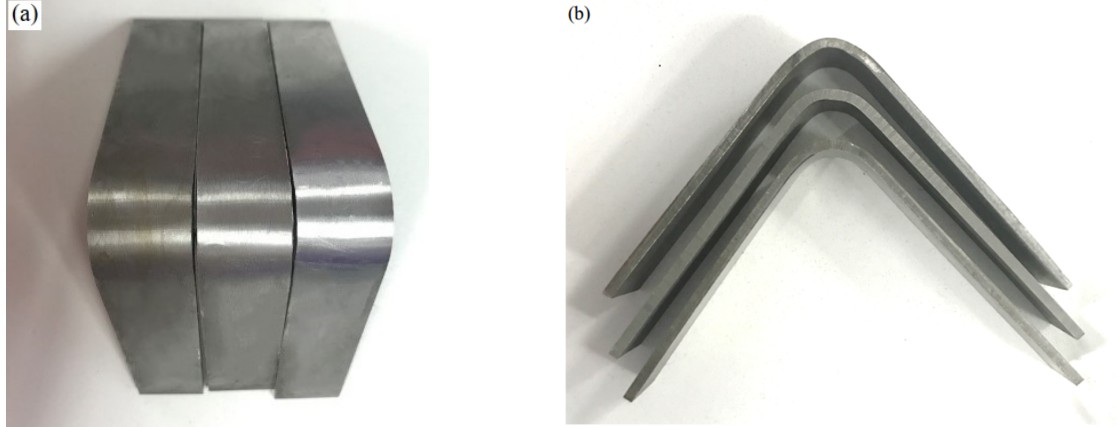

**Figure 7.** Surface situation of cold bending samples of the joint. (**a**) surface; (**b**) side.

### 3.4. Effect of Microstructure on Hardness of Welded Joint

Table 4 shows the hardness of BF and GB of the weld joint. The average hardness of BF in WS is 243.7 HV, which is higher than that of GB in CGZ.

**Table 4.** Hardness of BF and GB of the welded joint.

| Microstructure | BF | | | GB | | |
|---|---|---|---|---|---|---|
| | 1 | 2 | 3 | 1 | 2 | 3 |
| Hardness/HV | 240.3 | 242.1 | 248.7 | 238.9 | 232.3 | 236.4 |
| Average value/HV | 243.7 | | | 235.9 | | |

The hardness of WS was the highest because of the existence of M and BF (Figure 8a) and the hardness of CGZ was only inferior to WS with microstructure of AF and GB (Figure 8b). The main microstructure of FGZ and BM was F, so the hardness is lower. As shown in Figure 8c,d, there are higher dislocation density and smaller grain size of ferrite in FGZ compared with BM, making it macroscopically manifest as high strength and hardness because of more grain boundaries in unit area, greater hindrance to dislocation movement and greater resistance to material deformation. Therefore, the hardness of FGZ is greater than BM. Since the hardness of BM is the lowest, the welded joint is more likely to undergo plastic deformation at BM during stretching, which eventually leads to fracture.

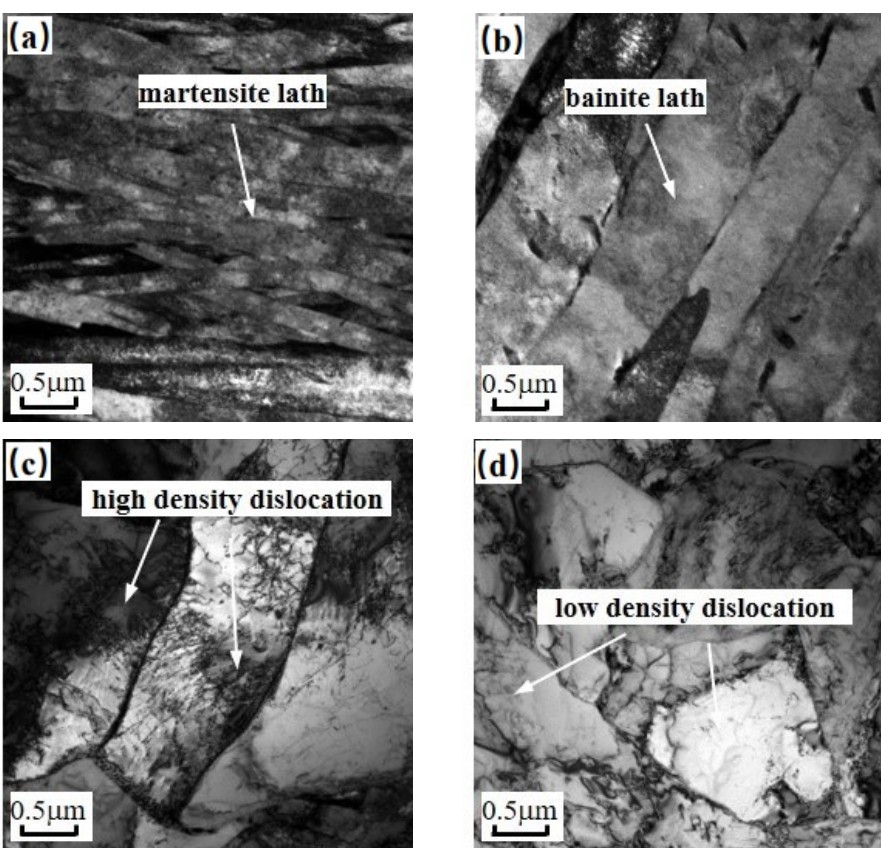

**Figure 8.** TEM micrographs of different regions of the welded joint. (**a**) WS; (**b**) CGZ; (**c**) FGZ; (**d**) BM.

*3.5. Effect of Microstructure on Impact Toughness of Welded Joint*

Figure 9 shows EBSD analysis results of each region of the welded joint. In EBSD analysis, the orientation difference of 0°–15° is defined as small angle grain boundary (shown in red line), which contains inclined grain boundaries and torsional grain boundaries, and the orientation difference is 15°–180° is defined as large angle grain boundary (shown in blue line). The small angle grain boundary energy is the energy of the dislocation group, while the large angle grain boundary energy is dominated by the core energy, and the energy value is between 0.15~1.2 J/m$^2$, which is independent of the orientation difference and is basically a constant value.

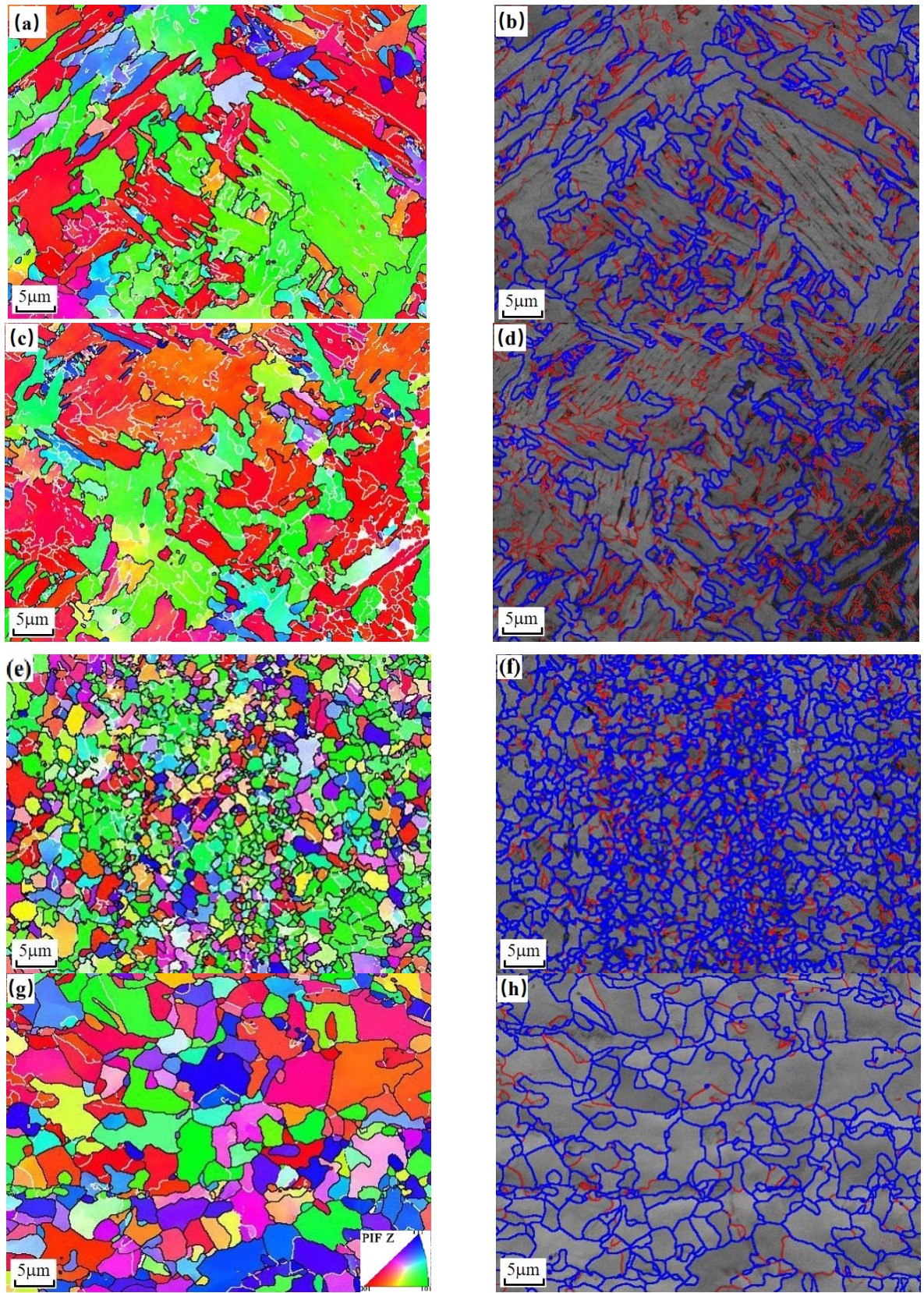

**Figure 9.** IPF and distribution of grain orientation of EBSD analysis results. (**a**,**b**) WS; (**c**,**d**) CGZ; (**e**,**f**) FGZ; (**g**,**h**) BM.

Figure 10 shows the results of small and large angle grain boundaries proportion in different regions of the welded joint. Red and blue in the histogram represent small angle grain boundaries and large angle grain boundaries, respectively. Because of the smallest grain boundary density and the existence of F-P banded structure, the impact toughness of BM is the worst, even though the percentage of large angle grain boundary in BM is relatively high [16,17]. The ferrite with the highest grain boundary density and percentage of large angle grain boundary (81.4%) in FGZ is fine. Due to the fine-grain strengthening mechanism, the impact toughness of FGZ is the best. Meanwhile, the fine blocky M/A island with dispersion distribution in FGZ is beneficial to the improvement of toughness. The percentage of large angle grain boundaries in WS and CGZ is 50.9% and 58.1%, respectively. Therefore, the impact toughness of CGZ and WS is in the middle. Because of the existence of AF and GB, the original austenite grains are divided into many areas with different orientations and uneven sizes. The growth of lath bundles with similar orientation and small orientation difference is limited by the region segmentation, which refines grains in the process of phase transition and increases the impact energy [18,19]. The morphology of M/A islands in GB are short bars or blocks, which are more effective in hindering the crack propagation than long strip of M/A islands in WS [20].

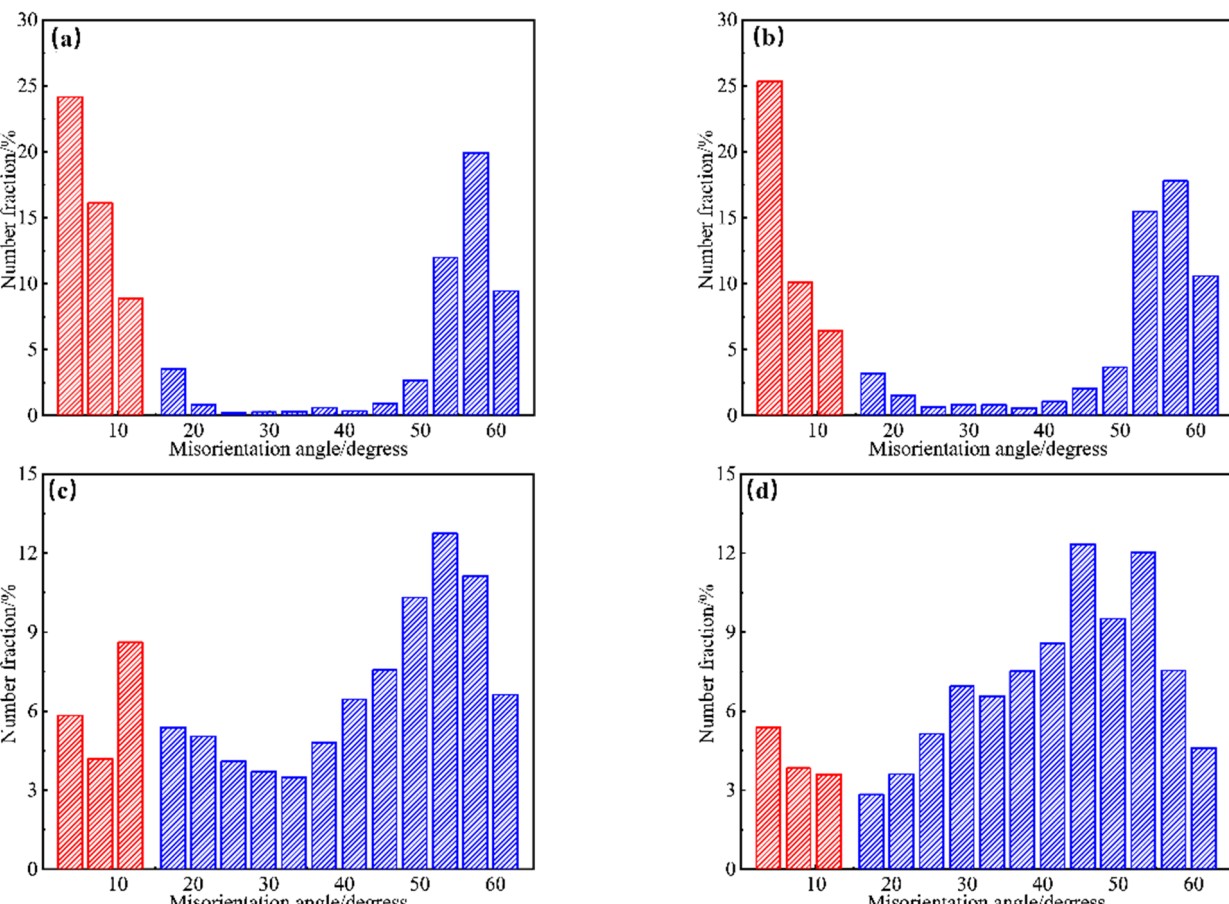

**Figure 10.** Results of small and large angle grain boundaries proportion. (**a**) WS; (**b**) CGZ; (**c**) FGZ; (**d**) BM.

## 4. Discussion

The welding heat affected zone (HAZ) is an important area that affects the comprehensive performance of the joint. Due to the inconsistent heating in this area and the uneven distribution of internal microstructure and properties, the coarsening of HAZ grains will lead to local softening, deterioration of joint strength and toughness, and reduction of joint bearing capacity. Flash butt welding is accompanied by the generation of reinforce-

ment. Although reinforcement can increase the strength of the joint to a certain extent, the sudden change of geometric shape will produce large stress concentration and various welding defects. In the actual production of wheels, reinforcement will generally be polished, but excessive weld reinforcement will also lead to metal waste and increase costs. Therefore, obtaining welded joints with small reinforcement and HAZ width and excellent comprehensive mechanical properties is the premise to ensure the welding performance of wheels.

Ferrite transformation has a certain influence on pearlite transformation. Ferrite nucleation and growth is essentially a process of carbon diffusion and migration from austenite. The larger the ferrite grain size is, the greater the carbon diffusion and migration during the nucleation and growth process. The higher the carbon concentration near the ferrite grain boundary, the more carbon-rich undercooled austenite transforms into pearlite during the subsequent cooling process. The grain size is small and the carbon content diffused during ferrite nucleation is low, which cannot reach the critical carbon concentration required for pearlite formation. Therefore, it is distributed along the pearlite grain boundary in the form of strip or granular carbide.

The energy of large angle grain boundary can be much higher than that of small angle grain boundary, which is equivalent to more atoms on large angle grain boundary deviating from the equilibrium position. When the crack expands to the large angle grain boundary, the crack occurs many times when crossing the large angle grain boundary because of the irregular arrangement of atoms, which consumes the crack propagation energy. Therefore, large angle grain boundaries have a better ability to hinder crack propagation compared with small angle grain boundaries [21,22].

Figure 11 shows the crack propagation path of acicular ferrite. The grain boundary of ferrite laths in same cluster is small angle grain boundary, while adjacent laths show different orientations. Large angle grain boundaries between adjacent laths become obstacles to crack propagation, which increases the impact energy [23,24]. FSP is the product of high temperature stage of solid phase transition in WS and the M in WS belong to brittle hard phase, which may cause the deterioration of impact toughness. Therefore, the toughness of CGZ is better than WS in terms of the microstructure and grain orientation.

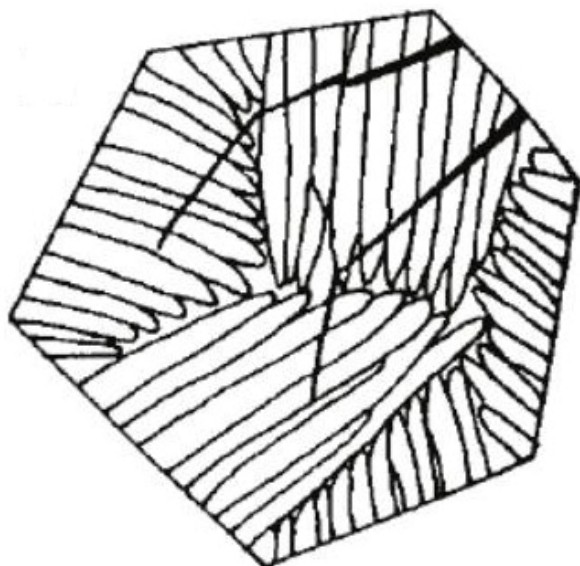

**Figure 11.** Crack propagation path in acicular ferrite.

Microstructure is the main factor affecting the strength and hardness of the joint. In welding thermal cycle, the joint is completely austenitized because of rapid heating to peak temperature, and then cooled to room temperature at different cooling rates, causing different types of phase transition. Carbon supersaturation in structure directly affects

hardness. Both ferrite transformation and pearlite transformation are diffusive phase transitions. The diffusion of Fe and C leads to lowest carbon supersaturation and hardness. The martensitic transformation belongs to non-diffusion phase transition, and all the carbon in the original austenite is retained in martensite, with the highest carbon supersaturation and hardness. In the process of bainite phase transition, some carbon atoms are diffused, and the hardness is in the middle [25].

Naylor made a thorough analysis of effective grains and propagation resistance of crack through the boundary of effective grains, as described by Equation (3) [26].

$$\sigma = \left[\frac{1.4Ea_cW}{Hd}\right]^{1/2} \tag{3}$$

where $E$ is elasticity modulus, $a_c$ is critical crack size, $W$ is the deflection plastic work on the lath, $H$ is the lath bundle width, and $d$ is the lath width. According to Equation (1), the crack propagation resistance is inversely proportional to the effective grain size $H^{-1/2}$, which is equivalent to toughness. Figure 12 shows the effective grain size of each region of the joint according to EBSD analysis results. The effective grain size of WS, CGZ, FGZ and BM is 3.31, 2.45, 1.98, and 4.5 μm, respectively. The effect of effective grain size on toughness can be summarized in two aspects: (1) the smaller the grain size, the larger the area of grain boundaries hindering the crack propagation, and the smaller the dislocation pile-up group in grain boundaries, which reduces the stress concentration. (2) The smaller the grain size, the higher the density of grain boundaries, and the lower the polymerization concentration of impurity elements on the boundary, which avoids the brittle fracture along grains. Therefore, low temperature impact toughness can be effectively improved by reducing effective grain size.

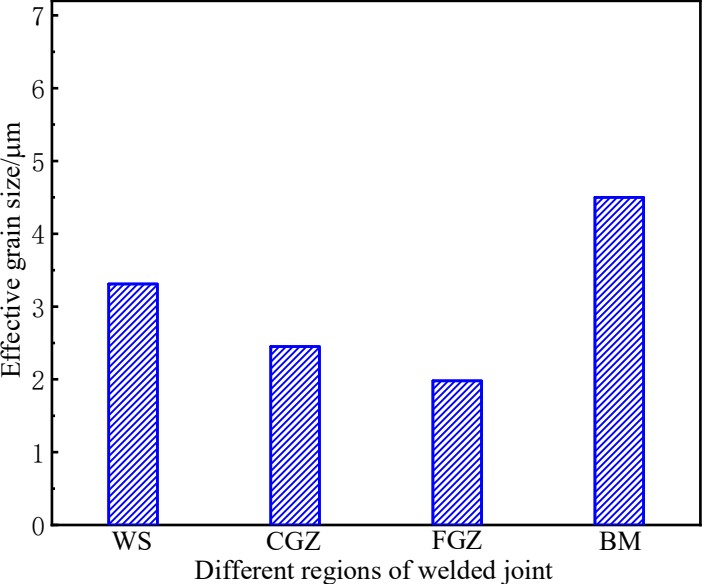

**Figure 12.** Effective grain size of each region.

In terms of the effect of effective grain size, the change of low temperature impact toughness of the welded joint is: FGZ > CGZ > WS > BM, which is consistent with the experimental results.

## 5. Conclusions

(1) The newly developed 590 MPa V-N microalloyed wheel steel had superior weldability under these conditions: flash current 48°/582.0 A, upset current 44°/516.6 A, workpiece gap of 1.5 mm. The tensile specimen of welded joint broke in BM and the

average tensile strength and elongation was 651 MPa and 20.7%, respectively. The hardness of joint gradually decreased from WS to BM and the hardness average value of WS, CGZ, FGZ, and BM were 280.9, 262.7, 232.9, and 217.3 HV, respectively. There was no softening tendency in the welded joint, and mechanical properties of the joint were excellent.

(2) The standard impact energy value of WS, CGZ, FGZ, and BM at $-40$ °C was 116, 128, 144, and 88 J, respectively. The change in low temperature impact toughness of welded joint followed the sequence: FGZ > CGZ > WS > BM.

(3) The microstructure of welded joint evolved in this order: welding seam (ferrite side plate + acicular ferrite + martensite)→coarse grain zone (acicular ferrite + granular bainite)→fine grain zone (fine grain ferrite + M/A island)→base metal (equiaxed ferrite + pearlite).

**Author Contributions:** Conceptualization, L.D. (Linxiu Du), K.C. and H.L.; methodology, C.G. and K.C; formal analysis, C.G. and T.L.; investigation, C.G., C.C. and Y.M.; data curation, C.C.; writing—original draft preparation, L.D. (Linxiu Du) and C.G.; writing—review and editing, L.D. (Linxiu Du), C.G. and X.G. All authors have read and agreed to the published version of the manuscript.

**Funding:** This research received no external funding.

**Institutional Review Board Statement:** The study did not require ethical approval.

**Informed Consent Statement:** Informed consent was obtained from all subjects involved in the study.

**Data Availability Statement:** Not applicable.

**Conflicts of Interest:** The authors declare no conflict of interest.

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
