# Peer review of "Microstructure Characteristics and Mechanical Properties of Flash Butt Welded 590 MPa V-N Microalloyed Heavy-Duty Truck Wheel Steel"

_metals, doi:10.3390/met13040688_

Round 1
Reviewer 1 Report
1. Authors should explain the meaning of the symbols used in the abstract. 2. In Table 1, lack of unit. 3. Do the same dependencies as for welding apply in the case of pressure welding? 4. Lack of explanation of BF symbol. 5. Is it really possible to distinguish between FSP, AF, or GB in the OM images, like the Authors did? 6. Are the Authors sure that in Fig. 1a and 1b there is more martensite than BF? 7. What do the Authors understand by "carbon emission"? 8. What is the size of M/A islands in the examined steel? 9. Lack of information how the notch was cut in the impact test specimens. 10. When applying the indenter load of 10 N, can we state that a given measurement corresponds to the hardness of BF or the hardness of GB? 11. How did the Authors distinguish between martensite lath and bainite lath in Figs. 4a and 4b?
12. Fig. 7 in the Reviewer's opinion does not bring anything to the article, it's redundant
Author Response
Dear editor:
We thank you very much for giving us an opportunity to revise our manuscript. We have substantially revised our manuscript after reading the comments provided by the reviewer and editor. We provided details in the revised version.
We have read reviewer's comments carefully and have made revision which was marked in red in the revised manuscript. We would like to express our great appreciation to you and reviewers for comments on our manuscript. Looking forward to hearing from you.
Thank you and best regards.
Yours sincerely.
Linxiu Du
Answers to reviewers:
First, we would like to thank the reviewers and the editor for positive and kind comments and suggestions.
[1] Authors should explain the meaning of the symbols used in the abstract.
Thanks for the reviewer's good evaluation and kind suggestion. The meaning of the symbols used in the abstract have been explain. WS, CGZ, FGZ and BM have been modified and replaced with “welding seam, coarse grain zone, fine grain zone and base metal " respectively.
[2] In Table 1, lack of unit.
Thanks for the reviewer's good evaluation and kind suggestion. We add unit to Table 1, and the revised table note is "Table 1 Chemical compositions of experimental steel (wt.%) ".
[3] Do the same dependencies as for welding apply in the case of pressure welding?
Thanks for the reviewer's good evaluation and kind suggestion. Pressure welding is a type of welding. So it has the same dependencies as for welding apply in the case of pressure welding.
[4] Lack of explanation of BF symbol.
Thanks for the reviewer's good evaluation and kind suggestion. BF has been modified and replaced with “bainite ferrite (BF) ".
[5] Is it really possible to distinguish between FSP, AF, or GB in the OM images, like the Authors did?
Thanks for the reviewer's good evaluation and kind suggestion. Ferrite-side-plate (FSP) from the austenite grain boundary proeutectoid ferrite side to plate-like growth to the crystal, from the morphological point of view, like pickaxe teeth. Acicular ferrite (AF) is distributed in the original austenite crystal, needle-shaped. Granular bainite (GB) generally appears granular. Therefore, the three are distinguishable.
[6] Are the Authors sure that in Fig. 4a and 4b there is more martensite than BF?
Thanks for the reviewer's good evaluation and kind suggestion. From multiple OM fields of view with different multiples, it can be effectively determined that there are few areas matching BF features, while martensite covers a wide range and has a large number. Combined with the SEM results of the field of view, there are more martensites and less BF content.
[7] What do the Authors understand by "carbon emission"?
Thanks for the reviewer's good evaluation and kind suggestion. Carbon emission refers to the process of human production and business activities to emit greenhouse gases to the outside world. The carbon emission process of austenite mentioned in the discussion is intended to express the process of carbon diffusion and migration from austenite. In order to avoid ambiguity, we modify the sentence and present a new sentence in the paper (marked in red).
The revised text is ‘Ferrite nucleation and growth is essentially a process of carbon diffusion and migration from austenite. The larger the ferrite grain size is, the greater the carbon diffusion and migration during the nucleation and growth process.’
[8] What is the size of M/A islands in the examined steel?
Thanks for the reviewer's good evaluation and kind suggestion. Because the content of M/A island is relatively small and the morphological difference is large, its size is not measured.
[9] Lack of information how the notch was cut in the impact test specimens.
Thanks for the reviewer's good evaluation and kind suggestion. The revised sentence is " The impact test was conducted on a 250HV fully digital instrumented pendulum impact tester using standard V-notch Charpy impact specimens (10×10×55 mm) according to ASTM A370."
[10] When applying the indenter load of 10 N, can we state that a given measurement corresponds to the hardness of BF or the hardness of GB?
Thanks for the reviewer's good evaluation and kind suggestion. When applying the indenter load of 10 N, we can state that a given measurement corresponds to the hardness of BF or the hardness of GB. Normally, When loading 10N, the surface is square and concave, and the equipment can measure the size and depth to give the hardness value.
[11] How did the Authors distinguish between martensite lath and bainite lath in Figs. 8a and 8b?
Thanks for the reviewer's good evaluation and kind suggestion. According to the TEM results, martensite lath and bainite lath can be distinguished. First, the position of WS and CGZ is different, and the phase composition and element content are different. Under the conditions of different cooling temperature and speed, the obtained microstructure will be significantly different, and martensite and bainite will be formed respectively. In addition, bainite is a mechanical mixture of carbon supersaturated ferrite and carbide. Generally, the upper bainite is relatively coarse and the carbon supersaturation is low. Martensite is a supersaturated solid solution of carbon in α-Fe, with high strength and hardness. It contains a large number of unevenly distributed dislocations in the form of lath and parallel bundles
[12] Fig. 7 in the Reviewer's opinion does not bring anything to the article, it's redundant
Thanks for the reviewer's good evaluation and kind suggestion. The figure has been adjusted. In the discussion part, it has a certain role in explaining the crack propagation path of acicular ferrite

Reviewer 2 Report
High demands are made on the quality of microalloyed heavy-duty truck wheel steel, especially in today's conditions of a steady increase in traffic volume, speed and axle loads. In connection with the above, the goal of the reviewed article is to increase the service life of welded microalloyed heavy-duty truck wheel steels and their operational reliability by reducing the likelihood of welding defects, increasing the structural strength of welded joints from new steel grades and eliminating the likelihood of operational defects. Therefore, the article is very important for science and practice, but there are several questions:
1. As a rule, when analysing the macro- and micro-mechanisms of destruction of Charpy samples, it is necessary to show the fracture surface of the specimens (macro level), and SEM. Figure 3 shows images of a fracture. But it is impossible to determine in which regions of fracture surface of Charpy specimens was SEM analysis is not possible. I recommend reading the article:
https://www.sciencedirect.com/science/article/abs/pii/S0167844215301981,
in it in Fig. 5 shows the failure surface research scheme. I propose to use it for notation in this article.
2. Perhaps in fig. 8 should also show the variance of effective grain size parameter. After all, its value will be certain boundaries, which is important for metallography and flaw detection.
3. It would be helpful to show the appearance of the seam. It would be useful for practical use.
4. How to use the obtained results to improve the welding technology?
Author Response
Dear editor:
We thank you very much for giving us an opportunity to revise our manuscript. We have substantially revised our manuscript after reading the comments provided by the reviewer and editor. We provided details in the revised version.
We have read reviewer's comments carefully and have made revision which was marked in red in the revised manuscript. We would like to express our great appreciation to you and reviewers for comments on our manuscript. Looking forward to hearing from you.
Thank you and best regards.
Yours sincerely.
Linxiu Du
Answers to reviewers:
First, we would like to thank the reviewers and the editor for positive and kind comments and suggestions.
[1] As a rule, when analysing the macro- and micro-mechanisms of destruction of Charpy samples, it is necessary to show the fracture surface of the specimens (macro level), and SEM. Figure 3 shows images of a fracture. But it is impossible to determine in which regions of fracture surface of Charpy specimens was SEM analysis is not possible. I recommend reading the article: https://www.sciencedirect.com/science/article/abs/pii/S0167844215301981,
in it in Fig. 5 shows the failure surface research scheme. I propose to use it for notation in this article.
Thanks for the reviewer's good evaluation and kind suggestion. I will read this article carefully, understand and apply it.
[2] Perhaps in fig. 8 should also show the variance of effective grain size parameter. After all, its value will be certain boundaries, which is important for metallography and flaw detection.
Thanks for the reviewer's good evaluation and kind suggestion. The variance of the effective grain size parameter is adjusted and displayed initially
[3] It would be helpful to show the appearance of the seam. It would be useful for practical use.
Thanks for the reviewer's good evaluation and kind suggestion. the appearance of the seam is shown in Fig. 2.
[4] How to use the obtained results to improve the welding technology?
Thanks for the reviewer's good evaluation and kind suggestion. The law and mechanism of flash welding will be further explored to facilitate large-scale industrial production.

Reviewer 3 Report
The author presents an experimental study about the weldability of a new microalloyed steel with V and N, for application, for example, in wheel steel
The work is well written and fluid, telling the reason for the study, the good weldability and the application of the material.
It is an absolutely normal study, with nothing particularly new, apart from the fact that it is a new steel, therefore worthy of publication.
In the annex I indicate small notes.

Author Response
Dear editor:
We thank you very much for giving us an opportunity to revise our manuscript. We have substantially revised our manuscript after reading the comments provided by the reviewer and editor. We provided details in the revised version.
We have read reviewer's comments carefully and have made revision which was marked in red in the revised manuscript. We would like to express our great appreciation to you and reviewers for comments on our manuscript. Looking forward to hearing from you.
Thank you and best regards.
Yours sincerely.
Linxiu Du

Round 2
Reviewer 2 Report
The authors took into account all my recommendations and improved the scientific content of the article. Only one recommendation was not taken into account. I will try to explain its importance below:
The authors write: Fig. 6 Fracture morphology of different regions of the welded joint….
For me, this leads to the formation of 4 questions at once:
- from which zone of the weld, the samples were cut, the fractures of which is shown in Fig. 6 a,b,c,d. In the title of Fig. 6 this information is missing. Please, add it.
- from which fracture zone of the Charpy specimens are shown the micromechanisms of fracture? The classification of fracture zones for Charpy specimens is shown, for example, in the article: https://www.sciencedirect.com/science/article/abs/pii/S0167844215301981
- why are SEM images shown at such high magnification? After all, it characterizes the micromechanisms of fracture only in the local area of failure. The choice of such scale of research requires additional commentary and justification.
- it is known that the analysis of fracture micromechanisms only at one scale of SEM studies is uninformative (https://www.mdpi.com/1996-1944/12/13/2051). It was better to analyse the macro- and micromechanisms of fracture. But the authors, for some reason, did not do it.
Author Response
Dear editor:
We thank you very much for giving us an opportunity to revise our manuscript. We have substantially revised our manuscript after reading the comments provided by the reviewer and editor. We provided details in the revised version.
We have read reviewer's comments carefully and have made revision which was marked in red in the revised manuscript. We would like to express our great appreciation to you and reviewers for comments on our manuscript. Looking forward to hearing from you.
Thank you and best regards.
Yours sincerely.
Linxiu Du
Answers to reviewers:
First, we would like to thank the reviewers and the editor for positive and kind comments and suggestions.
[1] from which zone of the weld, the samples were cut, the fractures of which is shown in Fig. 6 a,b,c,d. In the title of Fig. 6 this information is missing. Please, add it.
Thanks for the reviewer's good evaluation and kind suggestion. The experimental steel was welded using different flash butt welding processes .The impact test was conducted on a 250HV fully digital instrumented pendulum impact tester using standard V-notch Charpy impact specimens (10×10×55 mm) according to ASTM A370. Take an impact sample from the butt weld area. The center is the butt welding part, and both ends are the base metal. Due to the large internal stress and relatively uneven microstructure at the welding site, fracture usually occurs at the welding site to determine its impact performance.
[2] from which fracture zone of the Charpy specimens are shown the micromechanisms of fracture? The classification of fracture zones for Charpy specimens is shown, for example, in the article: https://www.sciencedirect.com/science/article/abs/pii/S0167844215301981.
Thanks for the reviewer's good evaluation and kind suggestion. Added content to the original text: " Isoaxial dimples are distributed in the impact fracture morphology of different regions of the joint, and no fluvial cleavage fracture feature was found. Therefore, it can be determined that the fracture mode in each region belongs to ductile fracture." In addition, the original text also discusses crack propagation in combination with large and small angle grain boundaries.
[3] why are SEM images shown at such high magnification? After all, it characterizes the micromechanisms of fracture only in the local area of failure. The choice of such scale of research requires additional commentary and justification.
Thanks for the reviewer's good evaluation and kind suggestion. This multiple of SEM can clearly observe the fracture morphology characteristics. For example, in Figure 6d, inclusions, coarsened precipitates, or coarse particles such as CaO at the bottom of the dimple preferentially germinate microcracks.
[4] it is known that the analysis of fracture micromechanisms only at one scale of SEM studies is uninformative (https://www.mdpi.com/1996-1944/12/13/2051). It was better to analyse the macro- and micromechanisms of fracture. But the authors, for some reason, did not do it.
Thanks for the reviewer's good evaluation and kind suggestion. Added content to the original text: " Cavities or coarse particles (Fig.6d) fully expand and polymerize under tensile and shear stresses, resulting in fracture, presenting equiaxed dimples of relatively uniform size. The final separation occurs along the direction of the maximum shear force, resulting in overall shear fracture.”
